# Mechano-modulatory synthetic niches for liver organoid derivation

Giovanni Sorrentino[1,5], Saba Rezakhani[2,5], Ece Yildiz[1], Sandro Nuciforo [3], Markus H. Heim [3,4], Matthias P. Lutolf [2✉] & Kristina Schoonjans [1✉]

The recent demonstration that primary cells from the liver can be expanded in vitro as organoids holds enormous promise for regenerative medicine and disease modelling. The use of three-dimensional (3D) cultures based on ill-defined and potentially immunogenic matrices, however, hampers the translation of liver organoid technology into real-life applications. We here use chemically defined hydrogels for the efficient derivation of both mouse and human hepatic organoids. Organoid growth is found to be highly stiffness-sensitive, a mechanism independent of acto-myosin contractility and requiring instead activation of the Src family of kinases (SFKs) and yes-associated protein 1 (YAP). Aberrant matrix stiffness, on the other hand, results in compromised proliferative capacity. Finally, we demonstrate the establishment of biopsy-derived human liver organoids without the use of animal components at any step of the process. Our approach thus opens up exciting perspectives for the establishment of protocols for liver organoid-based regenerative medicine.

[1] Laboratory of Metabolic Signaling, Institute of Bioengineering, School of Life Sciences and School of Engineering, Ecole Polytechnique Fédérale de Lausanne, 1015 Lausanne, Switzerland. [2] Laboratory of Stem Cell Bioengineering, Institute of Bioengineering, School of Life Sciences and School of Engineering, École Polytechnique Fédérale de Lausanne (EPFL), 1015 Lausanne, Switzerland. [3] Department of Biomedicine, University Hospital Basel, University of Basel, 4031 Basel, Switzerland. [4] Clinic of Gastroenterology and Hepatology, University Hospital Basel, University of Basel, 4031 Basel, Switzerland. [5] These authors contributed equally: Giovanni Sorrentino, Saba Rezakhani. ✉email: matthias.lutolf@epfl.ch; kristina.schoonjans@epfl.ch

Although the liver has a remarkable regenerative potential, chronic inflammation and scarring severely impair liver regeneration[1], making organ transplantation the only treatment option for patients with severe liver failure[2]. This therapeutic approach, however, is limited by the lack of liver donors, emphasizing the urgent need for cell-based therapies[3]. A promising alternative to liver transplantation comes from the recent breakthrough that liver organoids can be generated in vitro within animal-derived matrices (e.g. Matrigel) from mouse and human bile duct-derived bipotential facultative progenitor cells[4–6] or primary hepatocytes[4,5,7–9]. The first type of organoids are largely composed of progenitor cells that are genetically stable and can be differentiated into functional hepatocyte-like cells, which are able to engraft and increase survival when transplanted in a mouse model of liver disease[4,5]. However, the batch-to-batch variability of the three-dimensional (3D) matrices currently used for organoid derivation, as well as their mouse-tumour-derived origin, makes them unsuited for therapeutic ends. Recent work has suggested that composite matrices of fibrin and laminin-111, optimized for intestinal organoid culture, could also be used for liver organoid growth[10,11]. Owing to the mouse-tumour-derived laminin, these matrices are, however, incompatible with clinical use, and to the best of our knowledge there is no protocol available to expand and differentiate clinical-grade hepatic organoids[12,13].

In this study, we report the establishment of a chemically defined and mechano-modulatory 3D culture system for mouse and human hepatic progenitors and organoids for basic research and regenerative medicine applications. We optimized the efficiency of liver organoid derivation by tuning the mechanical properties of the synthetic microenvironment to match the physiological stiffness of the mouse liver. Finally, we accurately modelled the stiffness of the fibrotic liver, and demonstrate that aberrant liver mechanics negatively impact liver progenitor proliferation.

## Results

### Generation of a PEG-based synthetic niche for liver organoid culture.
We previously reported chemically defined 3D matrices for intestinal stem cell culture and organoid derivation[12], identifying design principles that could be adopted to mimic stem cell niches from different tissues. Here, we sought to develop a synthetic matrix for the efficient proliferation of liver progenitor cells by recapitulating key physical and biochemical characteristics of the hepatic microenvironment as an alternative to the established natural matrices Matrigel and collagen[4,14] (Supplementary Fig. 1a). To this aim, we first generated inert poly(ethylene glycol) (PEG) hydrogels enzymatically crosslinked by the activated transglutaminase factor XIIIa (FXIIIa)[15]. To mimic the mechanical properties of the mouse liver, we tuned the stiffness of PEG gels to physiological values (≈1.3 kPa)[16,17]. Key ECM proteins found in the native liver[18], such as laminin-111, collagen IV and fibronectin, were then incorporated in the PEG network, and soluble factors found in the hepatic niche, such as hepatocyte growth factor (HGF)[19], the Wnt agonist R-Spondin[4,20] and fibroblast growth factor 10 (FGF10)[21], were added to the culture medium, referred to as expansion medium (EM)[4].

Single dissociated mouse liver progenitor cells derived from Matrigel-expanded liver organoids were embedded into either Matrigel or PEG hydrogels and cultured in EM (Supplementary Fig. 1b). The functionalization of PEG hydrogels with fibronectin and laminin-111 led to efficient organoid generation, comparable to Matrigel (Fig. 1a, b). Replacement of the full-length fibronectin with its minimal integrin recognition peptide RGDSPG (Arg-Gly-Asp-Ser-Pro-Gly) led to similar results, suggesting that the

addition of a minimal adhesive moiety to the otherwise inert matrix is sufficient to promote extensive proliferation of liver progenitors (Fig. 1a, b). Cells expanding in PEG hydrogels modified with RGDSPG ('PEG-RGD') generated cystic structures characterized by a central lumen and a surrounding epithelium (Fig. 1c, e). Histology and gene expression analyses showed that these PEG-RGD-derived organoids possess a progenitor phenotype expressing stem/ductal markers such as Lgr5, Epcam, Krt19 and Sox9 (Fig. 1d, e and Supplementary Fig. 1c) and, in terms of morphology and gene expression, are indistinguishable from organoids grown in Matrigel (Fig. 1d, e). As expected, markers of fully differentiated hepatocytes such as Cyp3a11 were not expressed (Fig. 1d).

A major limitation of all current protocols for culturing epithelial organoids is an obligatory requirement of Matrigel (or similar natural ECM-derived matrices) in the first step of organoid generation. To test whether mouse liver organoids can be established in synthetic matrices without any initial Matrigel culture step, biliary duct fragments were isolated from mouse liver and directly embedded in PEG-RGD hydrogels (Supplementary Fig. 1b). Strikingly, after 6 days of culture, organoids emerged that could be serially passaged in culture (Fig. 1f and Supplementary Fig. 1d). PEG-RGD gels allowed organoid growth for more than 14 days (Supplementary Fig. 1e) without any significant structural deterioration. In contrast, Matrigel softened and no longer provided sufficient mechanical support already after 6 days of culture (Supplementary Fig. 1f), highlighting the importance of having a stably crosslinked matrix for long-term organoid culture.

### Liver organoids cultured in PEG hydrogel efficiently differentiate into hepatocyte-like cells.
Liver organoids can be differentiated in vitro into functional hepatocyte-like cells when cultured in the presence of specific differentiation medium (DM) containing inhibitors of Notch and TGF-β pathways[4,9] (Supplementary Fig. 2a). To test whether organoids derived in the synthetic matrix preserved the capacity for differentiation and hepatocyte maturation, we grew organoids in PEG-RGD gels and replaced the expansion medium with differentiation medium after 6 days, and analysed the expression of differentiated hepatocyte markers after 12 days (Supplementary Fig. 2d)[4]. Similar to Matrigel cultures, synthetic gels promoted a robust increase in the transcript levels of mature hepatocyte markers such as Cyp3a11, Alb, Ttr, Nr1h4 (Fxr), Slc2a2 (Glut2), Glul and Nr1h3 (Lxr) (Fig. 2a) and expression of ALB and HNF4α proteins (Fig. 2b), while markers of stem cells, such as Lgr5, disappeared (Supplementary Fig. 2b). As expected, during the differentiation process, cells acquired a characteristic hepatocyte-like morphology, as evidenced by a polygonal shape and expression of junction proteins such as ZO-1 and E-cadherin (Fig. 2c, d). Occasionally, hepatocyte-like cells showed poly-nucleation, a typical feature of hepatocytes (Fig. 2c, d and Supplementary Fig. 2e).

Next, to test whether hepatocyte-like cells generated in PEG-RGD hydrogels display hepatocyte-specific functions, we monitored albumin secretion. Liver organoids grown in DM showed a marked increase in albumin secretion, as compared to organoids cultured in EM (Fig. 2e). Moreover, hepatocyte-like cells were able to produce and secrete urea, as well as to accumulate glycogen, two enzymatically regulated processes that occur in mature hepatocytes (Fig. 2f, g). Moreover, the majority of differentiated cells were capable of internalizing low-density lipoproteins (LDL) from the culture medium, indicating that LDL receptor-mediated cholesterol uptake is functional (Fig. 2h). Finally, differentiation of organoids directly derived in PEG-RGD hydrogels was as efficient as in Matrigel (Supplementary Fig. 2c).

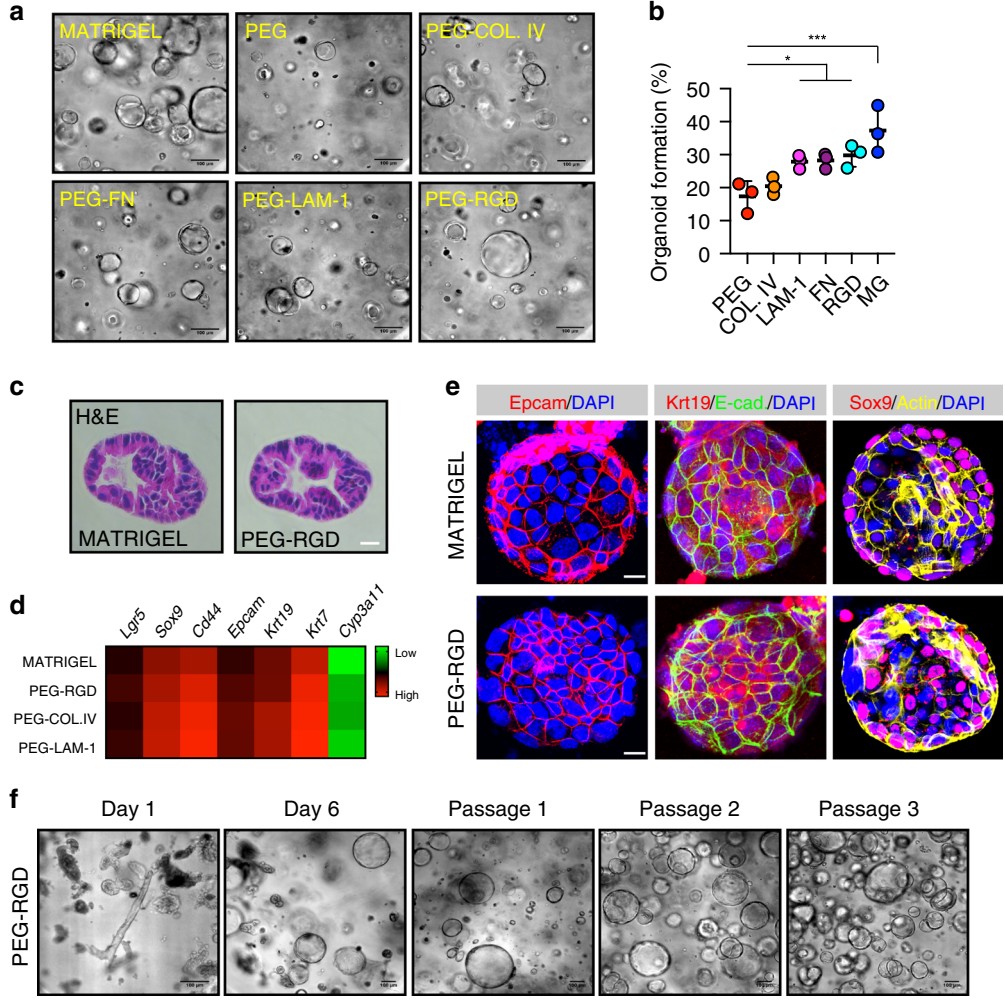

**Fig. 1 Liver organoid growth in Matrigel and PEG supplemented with ECM components. a** Mouse liver progenitor cells 3 days after embedding in: Matrigel (MG), plain PEG (PEG) and PEG functionalized with the indicated ECM components: COL.IV (collagen IV), FN (fibronectin), LAM-1 (laminin-1) and RGD-representing peptide (RGD). **b** Quantification of organoid formation efficiency relative to panel **a**. Graphs show individual data points derived from $n = 3$ independent experiments and means ± s.d., one-way Anova. *$P < 0.05$, ***$P < 0.001$. $P = 0.0321$; $P = 0.0266$; $P = 0.0119$; $P = 0.0003$. Source data are provided as a Source Data file. **c** Hematoxylin and eosin staining of Matrigel- and PEG-derived organoids. Scale bar 25 μm. **d** Gene expression was analysed by qRT-PCR in liver organoids 6 days after embedding in Matrigel and PEG hydrogels supplemented with different ECM factors. The heatmap represents ΔCt values as described in the method section. **e** Liver organoid immunostaining was performed 6 days after embedding of liver progenitor cells in Matrigel or PEG-RGD. **f** Liver organoids can be cultured in PEG hydrogels by directly embedding mouse biliary duct fragments. Representative pictures are shown. Micrographs (**c**, **e**, **f**) are representative of three independent experiments.

Altogether, these results demonstrate that liver organoids grown in PEG-RGD hydrogels can be readily differentiated into hepatocyte-like cells that mimic many of the established hepatic functions.

**Matrix stiffness controls liver organoid growth in an acto-myosin-independent manner.** Mechanical signals can play a critical role in controlling stem cell behaviour and tissue homeostasis[22], but also contribute to the manifestation of diseases[23–25]. Despite the recent progress in establishing novel 3D liver model systems[4,7,9], relatively little is known about the role of mechanics in regulating hepatic stem cell biology. This can be attributed to the fact that current culture systems rely on Matrigel, a matrix from tumour-derived origin, which has batch-to-batch stiffness variability and is not conducive to mechanical modification (Supplementary Fig. 3a). To test whether matrix mechanics affect liver organoid growth, we grew organoids in hydrogel of variable stiffness, ranging from values below the normal mouse liver stiffness (0.3 kPa) to those reaching physiological stiffness (1.3 kPa)[16,17].

Organoid formation efficiency and proliferation was profoundly affected by the mechanical properties of the matrix, with values mimicking physiological liver stiffness (between 1.3 and 1.7 kPa) being optimal (Fig. 3a, b and Supplementary Fig. 3b). Differentiation capacity, however, was unaffected by the degree of stiffness as induction of hepatic genes was maintained also when liver organoids were differentiated in soft gels (Supplementary Figs. 2d and 3c). These data demonstrate that optimizing the mechanical properties of the hydrogel represents a critical step in the efficient generation of liver organoids.

Given the pivotal role of the Hippo pathway nuclear effector, Yes associate protein (YAP), in the transduction of microenvironment mechanical cues downstream of integrins[26], we next tested whether YAP activation could potentially explain the observed matrix stiffness dependence. We monitored the expression of canonical YAP target genes and YAP subcellular localization in organoids cultured in soft (0.3 kPa) and physiologically stiff (1.3 kPa) matrices. Interestingly, in stiffer hydrogels, YAP target genes expression and nuclear accumulation were increased

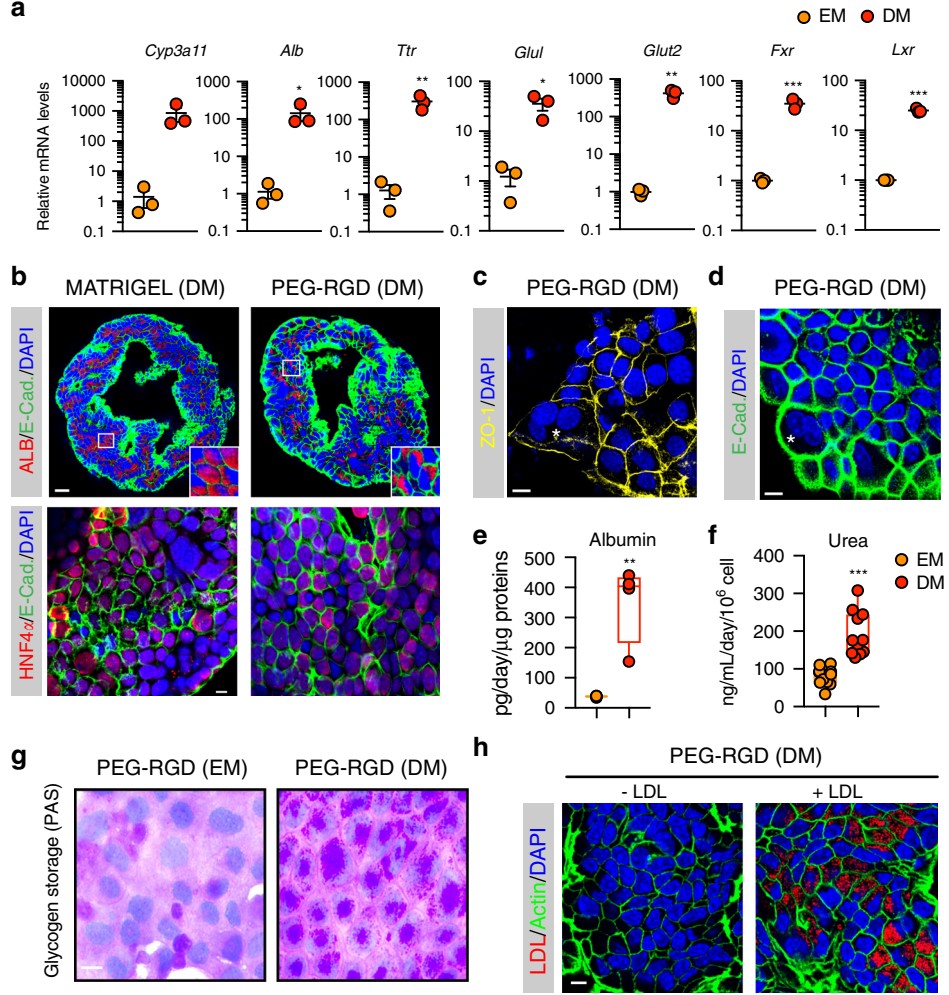

**Fig. 2 Differentiation of liver organoids into hepatocyte-like cells in PEG-RGD hydrogels. a** Gene expression was analysed by qRT-PCR in liver organoids maintained in expansion medium (EM) or differentiation medium (DM) in PEG-RGD hydrogels. Graphs show individual data points derived from $n = 3$ independent experiments and means ± SEM, unpaired Student's one-tailed t-test. *$P < 0.05$, **$P < 0.01$, ***$P < 0.001$ ($P = 0.030$; 0.0077; 0.0007; 0.0126; 0.0009; 0.00003). **b** Representative confocal immunofluorescence images of Albumin (ALB), Hnf4a and E-Cadherin (E-CAD.). Scale bars: 25 μm (top), 10 μm (bottom). **c** Representative confocal immunofluorescence images of ZO-1 in liver organoids maintained in differentiation medium (DM) in PEG-RGD hydrogels. The asterisk indicates a binucleated cell. Scale bar 10 μm. **d** Liver organoids maintained in differentiation medium (DM) in PEG-RGD hydrogels. E-cadherin was used to visualize cell borders. The asterisk indicates a binucleated cell. Scale bar 10 μm. **e** Albumin secretion was quantified in the supernatant of organoids embedded in PEG-RGD hydrogels and maintained in EM or DM. Center line, median; box, interquartile range (IQR); whiskers, range (minimum to maximum). Red and orange dots, individual data points derived from $n = 4$, unpaired Student's two-tailed t-test. **$P < 0.01$ ($P = 0.0031$). **f** Urea production was quantified in the supernatant of organoids embedded in PEG-RGD hydrogels and maintained in EM or DM. Center line, median; box, interquartile range (IQR); whiskers, range (minimum to maximum). Red and orange dots, individual data points derived from $n = 11$, unpaired Student's two-tailed t-test. ***$P < 0.001$ ($P < 0.0001$). **g** Glycogen accumulation was assessed by PAS (Periodic-Acid Schiff) staining in liver organoids embedded in PEG-RGD hydrogels and maintained in EM or DM. Scale bar 10 μm. **h** LDL uptake was monitored by Dil-ac-LDL fluorescent substrate in liver organoids maintained in DM in PEG-RGD hydrogels. Scale bar 10 μm. Micrographs (**b–d**, **g**, **h**) are representative of three independent experiments. Source data are provided as a Source Data file.

compared to soft matrices (Fig. 3c and Supplementary Fig. 3d). To examine whether an activated integrin/YAP signalling axis is functionally required for organoid derivation in physiologically stiff matrices, we treated organoid cultures with PF-573228 (PF), an inhibitor of the integrin effector focal adhesion kinase (FAK), or with the YAP inhibitor Verteporfin (VP)[27] (Fig. 3d). Both treatments prevented the increase in organoid formation induced in stiffer matrices (Fig. 3e and Supplementary Fig. 3e), indicating that the integrin/YAP module is required in coordinating growth of liver progenitors in response to mechanical stimuli.

We then sought to identify the other components of the FAK-YAP cascade that may play a role in modulating the stiffness response in our system. Since remodelling of the actin

cytoskeleton has been identified as a key event upstream of YAP activation[28–30], we assessed its putative involvement in physiological stiffness-induced organoid growth by inhibiting acto-myosin contractility with blebbistatin[31]. Surprisingly, blebbistatin treatment significantly enhanced organoid formation (Fig. 3f) with an efficiency comparable to ROCK inhibitors (Supplementary Fig. 3g), indicating that matrix stiffness promotes organoid growth independently of cytoskeletal dynamics, and that acto-myosin contractility rather interferes with normal liver progenitor expansion. However, tyrosine phosphorylation of YAP by the Src family of kinases (SFK), an alternative route for integrin-dependent and acto-myosin independent YAP activation[32–40], was increased by matrix

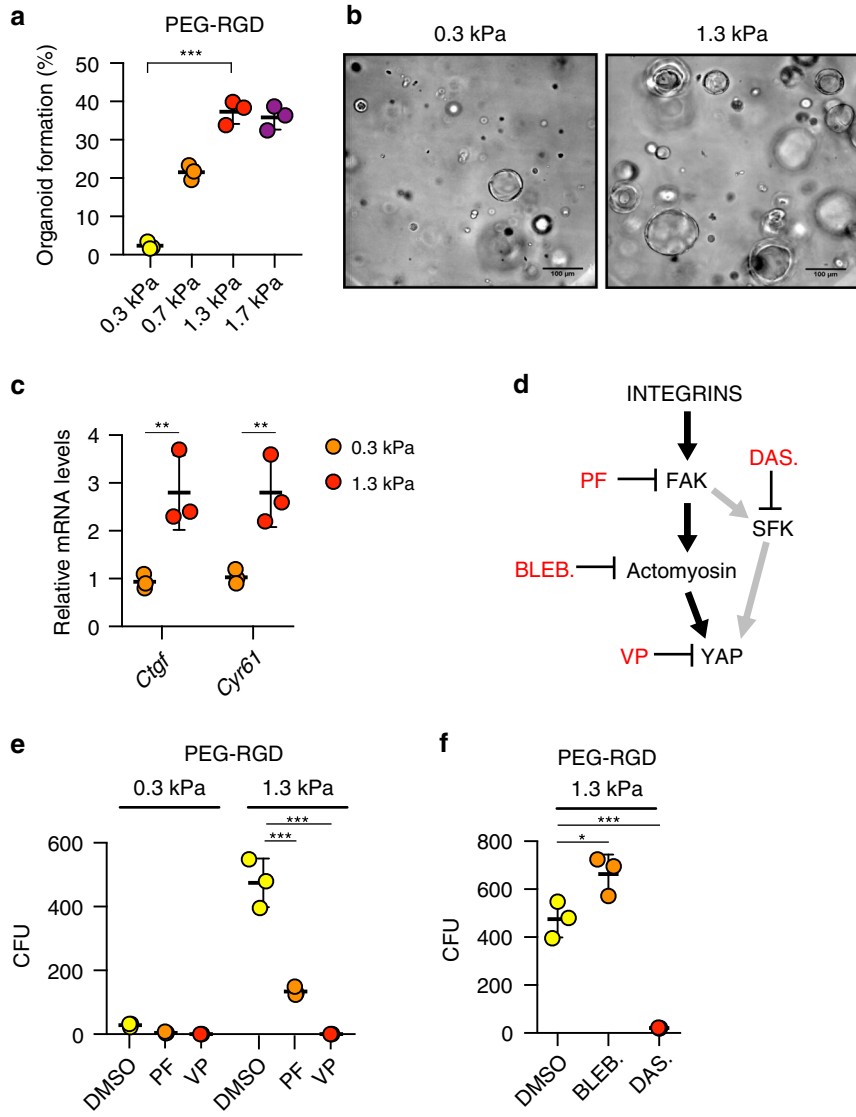

**Fig. 3 Effect of matrix stiffness on liver organoid formation. a** Effect of matrix stiffness on organoid formation efficiency. Graphs show individual data points derived from $n = 3$ independent experiments and means ± s.d. ($P < 0.0001$). **b** Representative image of organoids 3 days after embedding in PEG-RGD hydrogels of indicated stiffness. **c** Gene expression was analysed by qRT-PCR in liver organoids 6 days after embedding in PEG-RGD hydrogels with indicated stiffness. Graphs show individual data points derived from $n = 3$ independent experiments and means ± s.d. ($P = 0.0088$; 0.0077). **d** Schematic representation of cellular mechano-signalling pathways. Inhibitors of key elements are depicted in red. **e** Effect of indicated inhibitors on organoid formation efficiency in soft (300 Pa) and physiologically stiff (1.3 kPa) PEG-RGD hydrogels. CFU (colony forming unit). Graphs show individual data points derived from $n = 3$ independent experiments and means ± s.d. ($P < 0.0001$). **f** Effect of indicated inhibitors on organoid formation efficiency in physiologically stiff (1.3 kPa) PEG-RGD hydrogels. CFU (colony forming unit). Graphs show individual data points derived from $n = 3$ independent experiments and means ± s.d. ($P = 0.0198$; 0.0003). *$P < 0.05$, **$P < 0.01$, ***$P < 0.001$ one-way Anova (**a**, **f**) or two-way Anova (**c**, **e**). Source data are provided as a Source Data file.

stiffness (Supplementary Fig. 3f). Of interest, treatment with Dasatinib[41], a FDA-approved SFK inhibitor, fully abolished YAP phosphorylation and organoid growth in physiologically stiff (1.3 kPa) matrices (Fig. 3f and Supplementary Fig. 3f). These results corroborate the importance of the integrin/SFK/YAP signalling pathway in liver progenitor proliferation in response to differential mechanical inputs.

**PEG hydrogels can be tuned to model fibrotic liver mechanics**. Liver disease progression is strongly associated with abnormal tissue architecture and mechanotransduction[16,42,43]. Indeed, as a direct effect of aberrant ECM deposition in the fibrotic liver, tissue stiffness increases in time and severely compromises its function[44–46]. In fact, the changes in liver stiffness associated with

disease are used for diagnostics based on longitudinal noninvasive monitoring[47]. We reasoned that liver organoids grown in defined hydrogels recapitulating the stiffness of fibrotic liver could serve as physiologically relevant 3D model to investigate how stem cells translate aberrant mechanical inputs into disease-relevant phenotypes. To this aim, we generated fibrosis-mimicking hydrogels with a stiffness of 4 kPa[16,17]. Strikingly, these hydrogels led to a significant impairment of organoid formation (Fig. 4a, b and Supplementary Fig. 3a), demonstrating that an abnormal ECM stiffness is sufficient to decrease the liver progenitor proliferative capacity. In this condition liver organoids showed a reduction in the expression of hepatic progenitor markers (Fig. 4c) and upregulation of genes involved in cellular response to hepatic injury (Fig. 4d and Supplementary Fig. 4a), indicating impaired stemness potential and concomitant induction of a stress

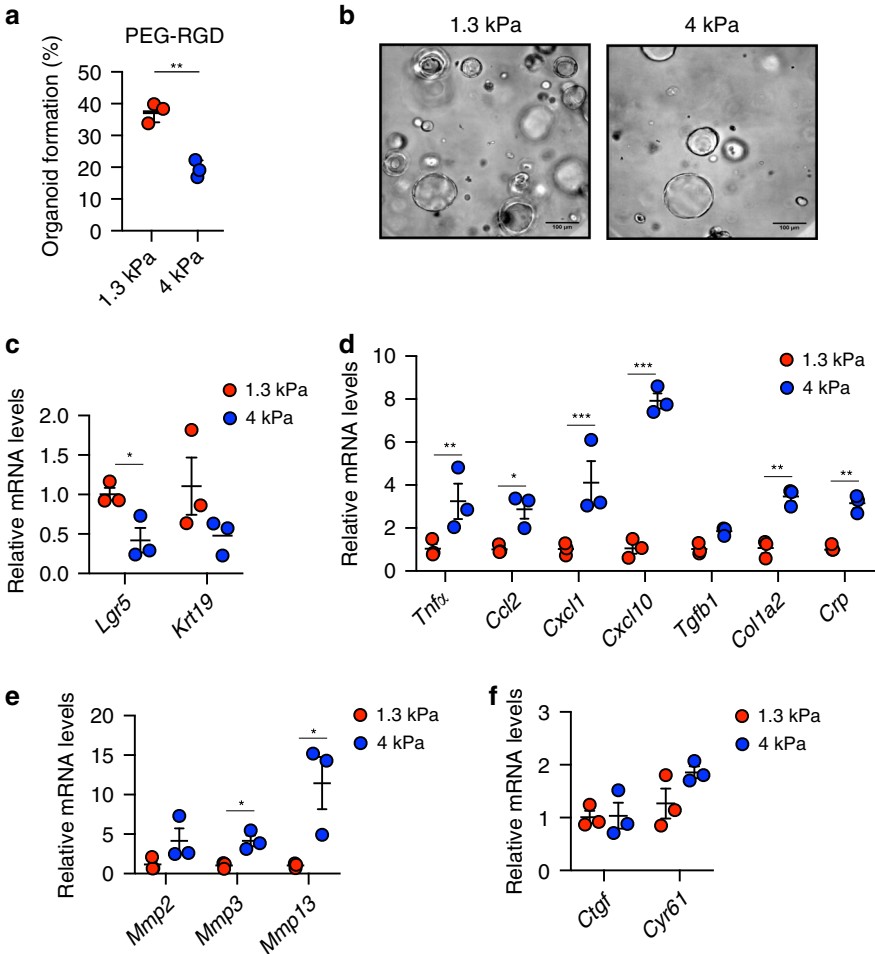

**Fig. 4 Hydrogels mimicking the stiffness of native fibrotic liver affect the growth of liver organoids and promote a stress response. a** Effect of matrix stiffness on organoid formation efficiency. Graphs show individual data points derived from $n = 3$ independent experiments and means ± SEM, unpaired Student's two-tailed $t$-test ($P = 0.0017$). **b** Representative image of organoids 3 days after embedding in PEG-RGD hydrogels of indicated stiffness. Scale bars: 100 μm. **c–f** Gene expression was analysed by qRT-PCR in liver organoids 6 days after embedding in PEG-RGD hydrogels with indicated stiffness. Graphs show individual data points derived from $n = 3$ independent experiments and means ± SEM. c ($P = 0.0218$); **d** ($P = 0.0051$; 0.0223; 0.0001; 0.0001; 0.0021; 0.0056); e (0.0127; 0.033). *$P < 0.05$, **$P < 0.01$ ***$P < 0.001$ unpaired Student's two-tailed $t$-test (**c**, **e**) or two-way Anova (**d**, **f**). Source data are provided as a Source Data file.

response. Finally, fibrosis-mimicking hydrogels led to an increase in the expression of matrix metalloproteases (Fig. 4e), a compensatory phenomenon known to be induced in response to increased ECM stiffness[48–50]. Surprisingly, in these conditions YAP activation was not affected (Fig. 4f), suggesting the existence of other pathways controlling liver progenitor growth in response to increased stiffness. These results suggest that synthetic hydrogels may be a useful tool to assess the contribution of mechanical inputs on liver diseases.

**PEG hydrogels allow derivation and culture of human liver organoids.** To test the potential clinical relevance of our findings, we assessed whether the PEG-RGD hydrogels, initially designed for expansion and differentiation of mouse liver organoids, were also suitable for culturing human organoids. We first generated Matrigel-derived organoids from human non-tumorigenic liver needle biopsies[5]. Human liver progenitor cells were then embedded in PEG-RGD and cultured in human expansion medium (HEM) (Supplementary Fig. 4b). In these conditions human liver progenitor cells generated organoids that could be expanded over multiple passages (Fig. 5a). Similar to mouse,

human organoid generation efficiency in PEG-RGD was comparable to Matrigel (Fig. 5a), and was critically dependent on changes in hydrogel stiffness (Fig. 5b). Moreover, when cultured in HEM, human liver organoids grown in PEG-RGD expressed progenitor cell markers, such as KRT19 (Fig. 5c), but readily differentiated into human hepatocyte-like cells when cultured in human differentiation medium (HDM) (Fig. 5d and Supplementary Fig. 4c).

Finally, in order to generate clinically relevant human liver organoids, we tested the possibility of establishing organoids from human patients without the interference of any animal-derived matrices. To this aim, freshly isolated liver biopsies from six patients were digested and directly embedded in PEG-RGD hydrogels (Fig. 5e). Strikingly, after 8 days of culture, organoid formation was scored from progenitor cells of all patients (Fig. 5f) and could be passaged (Fig. 5g) and differentiated (Supplementary Fig. 4d). Moreover, liver organoids could be frozen-thawed in PEG-RGD hydrogels (Supplementary Fig. 4e).

Altogether, these data provide proof-of-concept that human liver progenitor cells can be derived and maintained in vitro within synthetic matrices.

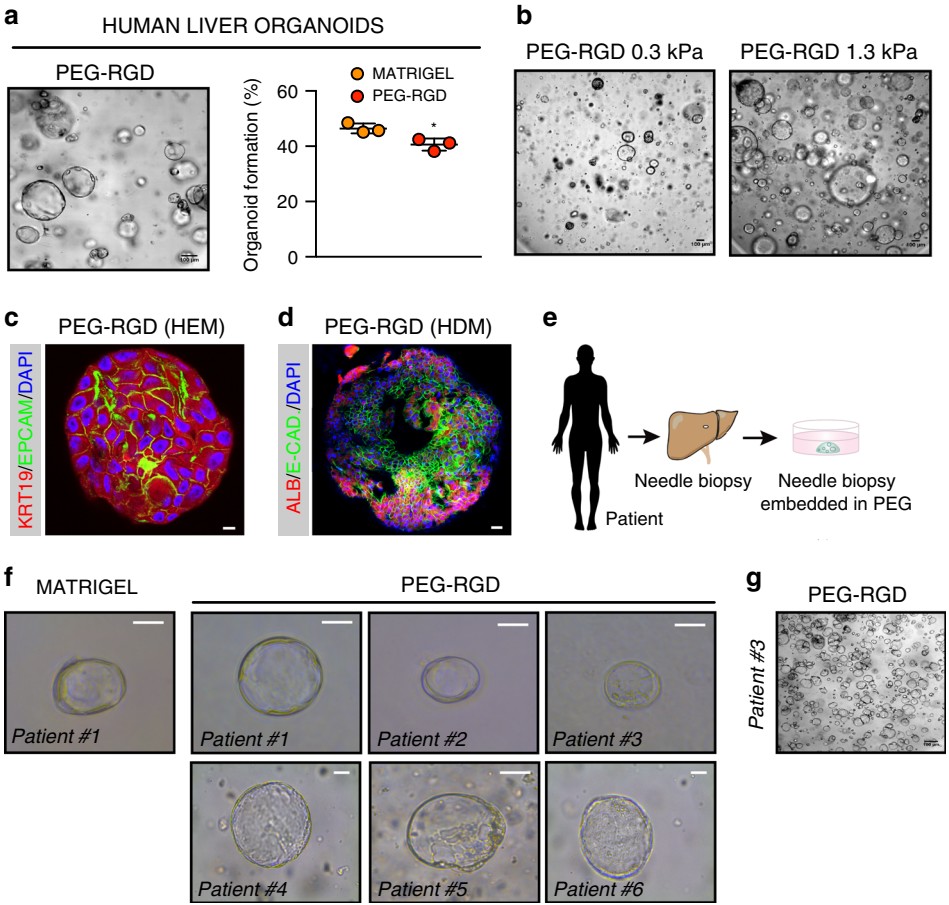

**Fig. 5 PEG-RGD hydrogels allow for establishment, expansion and differentiation of patient-derived human liver organoids. a** Matrigel-derived human liver progenitor cells were embedded in physiologically stiff PEG-RGD hydrogels and passaged in human expansion medium (HEM) with an efficiency comparable to Matrigel. Representative picture (left) and quantification of organoid formation efficiency (right). Graphs show individual data points derived from $n = 3$ independent experiments and means ± SEM, unpaired Student's two-tailed $t$-test. *$P < 0.05$ ($P = 0.0231$). **b** Effect of matrix stiffness on human liver organoid formation. **c** Representative confocal immunofluorescence image of KRT19 and Epcam in liver organoids embedded in physiologically stiff PEG-RGD hydrogels and cultured in HEM. Scale bar 10 μm. **d** Representative confocal immunofluorescence image of Albumin and E-Cadherin in liver organoids cultured in human differentiation medium (HDM), Scale bar 25 μm. **e, f** Freshly isolated liver biopsies from six patients were digested and directly embedded in PEG-RGD hydrogels. Scale bar 50 μm. **g** PEG-RGD-derived human liver progenitor cells were embedded in physiologically stiff hydrogels and passaged in human expansion medium (HEM). Scale bar 100 μm. Micrographs (**c, d, g**) are representative of three independent experiments. Source data are provided as a Source Data file.

## Discussion

We report here the establishment of a fully defined 3D culture system for mouse and human hepatic progenitors and organoids. We demonstrated that liver organoids can be expanded and maintained in such minimal environments with an efficiency that is comparable to Matrigel, but without its main disadvantages of structural instability, batch-to-batch variability and clinical incompatibility. By tuning the stiffness of the synthetic networks to match the physiological levels of the liver, we optimized the efficiency of liver organoid derivation, and identified integrin-SFK-YAP as a mechano-sensitive axis that is required for liver organoid growth. Interestingly, in contrast to intestinal organoids[12], we found that liver progenitor cells transduce mechanical signals in an acto-myosin independent manner and instead require activation of the tyrosine kinase Src to support epithelial tissue formation. Moreover, we used PEG hydrogels to accurately model the aberrant mechanical properties of the fibrotic liver, providing evidence that aberrant liver stiffness negatively impacts liver progenitor proliferation. This experimental setup provides a standardized framework to study hepatic progenitor cells in a defined mechanical environment, which may further our understanding of the underappreciated role of mechanical cues in modulating the molecular properties and signatures of this cell population in healthy and fibrotic liver. Finally, our data showing that clinically relevant human stem/progenitor cells can be grown in vitro without any requirement of animal-derived matrices, may open exciting perspectives for the establishment of protocols for liver organoid-based clinical applications.

## Methods

**Data reporting**. The experiments were not randomized and the investigators were not blinded to allocation during experiments and outcome assessment. No statistical methods were used to determine sample size. The experiments were repeated at least three times or with at least three different donors to control biological variations.

**Animals and ethical approval**. Liver tissues were harvested from 8–12-weeks-old euthanized C57BL/6 J male mice. All the animal experiments were authorized by the Veterinary Office of the Canton of Vaud, Switzerland under the license authorization no. 3263. The mice were housed in groups of five mice per cage with access to food and water. The temperature of the animal facility was set at 22 ± 2 °C, the hygrometry at 40–60% and the light cycle (12:12) from 7:00 to 19:00.

**Enzymatically crosslinked hydrogel precursor synthesis.** Hydrogel precursors were synthesized as previously reported[51]. Briefly, vinylsulfone functionalized 8-arm PEG (PEG-VS) was purchased from NOF. The transglutaminase (TG) factor XIII (FXIIIa) substrate peptides Ac-FKGG*GPQGIWGQ*-ERCG-NH2 with matrix metalloproteinases (MMPs)-sensitive sequence (in italics), Ac-FKGG-GDQGIAGF-ERCG-NH2, H-NQEQVSPLERCGNH2 and the RGD-presenting adhesion peptide H-NQEQVSPLRGDSPG-NH2, were purchased from GL Biochem. FXIIIa substrate peptides and 8-arm PEG-VS were dissolved in triethanolamine (0.3 M, pH 8.0) and mixed at 1.2 stoichiometric excess (peptide-to-VS group), and allowed to react for 2 h under inert atmosphere. The reaction solution was dialysed (Snake Skin, MWCO 10 K, PIERCE) against ultrapure water for 3 days at 4 °C, after which the products were lyophilized and dissolved in ultrapure water to make 13.33% w/v stock solutions.

**Formation and dissociation of PEG hydrogels.** PEG precursor solutions were mixed in stoichiometrically balanced ratios to form hydrogel networks of a desired final PEG content. Addition of thrombin-activated FXIIIa (10 U ml−1; Galexis) triggered the hydrogel formation in the presence of Tris-buffered saline (TBS; 50 mM, pH 7.6) and 50 mM CaCl2. The spare reaction volume was used for the incorporation of dissociated liver stem cells, fragments of liver bile ducts, and ECM components: RGD-presenting adhesion peptide, fibronectin (0.5 mg ml−1; R&D systems), laminin-111 (0.2 mg ml−1; Invitrogen), collagen IV (0.2 mg ml−1; BD Bioscience). Gels cast on PDMS-coated 24-well plate were allowed to crosslink by incubation at 37 °C for 10 min. To release the grown colonies for further processing, gels were detached from the bottom of the plates using a tip of a metal spatula and transferred to 15-ml Falcon tube containing 1 ml of Dispase (1 mg/ml, Thermo Fisher Scientific). After 10 minutes enzymatic digestion, the reaction was quenched using 10% FBS containing 1 mM EDTA, washed with cold basal medium and centrifuged for 3 min at 1000 rpm.

**Mechanical characterization of PEG hydrogels.** Elastic modulus (G') of hydrogels was measured by performing small-strain oscillatory shear measurements on a Bohlin CVO 120 rheometer with plate-plate geometry. Briefly, 1–1.4 mm thick hydrogel discs were prepared and allowed to swell in water for 24 h. The mechanical response of the hydrogels sandwiched between the parallel plates of the rheometer was recorded by performing frequency sweep (0.1–10 Hz) measurements in a constant strain (0.05) mode at 25 °C.

**Quantification of liver organoid formation efficiency.** Phase contrast z-stacks images were collected through the entire thickness of the PEG gels (every 15 μm) at four different locations within the gels (Nikon Eclipse Ti). The Cell Counter plugin in ImageJ (NIH) was used to quantify the percentage of single cells that formed colonies after 3 days of culture in expansion medium.

**Culture of mouse and human liver organoids.** Mouse liver organoids were established from biliary duct fragments as previously described with some modifications[4]. Briefly, liver tissues were digested in digestion solution (Collagenase type XI 0.012%, dispase 0.012%, FBS 1% in DMEM medium) for 2 h. When digestion was complete, bile ducts were pelleted by mild centrifugation (200 rpm for 5 min) and washed with PBS. Isolated ducts were then resuspended either in Type I Collagen (8–11 mg/ml prepared following manufacturer's instruction—from Corning), Matrigel (BD Bioscience) or PEG precursor solution and cast in 10 μl droplets in the centre of the wells in a 48-well plate. After the gels were formed, 250 μl of isolation medium was added to each well. Isolation medium was composed of AdDMEM/F-12 (Invitrogen) supplemented with B-27 and N-2 (both GIBCO), 1.25 μM N-acetylcysteine (Sigma–Aldrich), 10 nM gastrin (Sigma–Aldrich) and the following growth factors: 50 ng ml−1 EGF (Peprotech), 1 μg ml−1 Rspo1 (produced in-house), 100 ng ml−1 Fgf10 (Peprotech), 10 mM nicotinamide (Sigma–Aldrich), 50 ng ml−1 HGF (Peprotech), Noggin (100 ng ml−1 produced in-house), Wnt 3a (1 μg ml−1, Peprotech) and Y-27632 (10 μM, Sigma). After the first 4 days, isolation medium was changed with expansion medium (EM), which consists of isolation medium without Noggin, Wnt and Y-27632. One week after seeding, organoids were removed from the Matrigel or PEG hydrogel, dissociated into single cells using TrypLE express (Gibco), and transferred to fresh Matrigel or PEG hydrogels. Passaging was performed in 1:3 split ratio once per week. Plasmids for Rspo1 and Nog production were a kind gift from Joerg Huelsken. Liver progenitor cells were treated with the following compounds for YAP-inhibition experiments: Verteporfin (10 μM, Sigma), Dasatinib (10 μM, Selleckchem), Blebbistatin (10 μM StemCell Technologies), PF-573228 (10 μM Tocris).

**Human liver biopsies and generation of human organoids.** Human tissues were obtained from patients undergoing diagnostic liver biopsy at the University Hospital Basel. Written informed consent was obtained from all patients. The study was approved by the ethics committee of the northwestern part of Switzerland (Protocol Number EKNZ 2014-099). Ultrasound (US)-guided needle biopsies were obtained with a coaxial liver biopsy technique as described previously[50]. One biopsy cylinder was fixed in formalin and paraffin-embedded for histopathological diagnosis. Additional cylinders were collected in advanced DMEM/F-12 (GIBCO) for organoid generation. Patient clinical information are shown in Supplementary

Table 2. Human liver organoids were generated as previously described with some modifications[5,50]. Briefly, biopsies were placed in advanced DMEM/F-12 (GIBCO) and transported to the laboratory on ice. Liver samples were then digested to small-cell clusters in basal medium containing 2.5 mg/mL collagenase IV (Sigma) and 0.1 mg/mL DNase (Sigma) at 37 °C. Cell clusters were embedded in Matrigel or PEG gels, cast and after the gels were formed, human isolation medium (HIM) was added. HIM is composed of advanced DMEM/F-12 (GIBCO) supplemented with B-27 (GIBCO), N-2 (GIBCO), 10 mM nicotinamide (Sigma), 1.25 mM N-acetyl-L-cysteine (Sigma), 10 nM [Leu15]-gastrin (Sigma), 10 μM forskolin (Tocris), 5 μM A8301 (Tocris), 50 ng/mL EGF (PeproTech), 100 ng/mL FGF10 (PeproTech), 25 ng/ml HGF (PeproTech), 1 μg ml−1 Rspo1 (produced in-house), Wnt 3a (1 μg ml−1, Peprotech), Y-27632 (10 μM, Sigma) and Blebbistatin (10 μM StemCell Technologies). After the first 4 days, isolation medium was changed with human expansion medium (HEM), which consists of HIM without Noggin, Wnt and Y-27632.

**Mouse hepatocyte differentiation.** Single cells were seeded and kept for 6 days in EM. Then the medium was changed to differentiation medium (DM), which no longer contains Rspo1, HGF and nicotinamide and instead contains A8301 (50 nM, Tocris Bioscience) and DAPT (10 nM, Sigma–Aldrich). Cells were maintained in DM for 12 days. During the last 3 days DM was also supplemented with dexamethasone (Sigma, 3 μM). Medium was changed every 2 days.

**Human hepatocyte differentiation.** Single cells were seeded and kept 6 days in HEM. Then the medium was changed to differentiation medium (HDM), which no longer contains Rspo1, HGF and nicotinamide and instead contains A8301 (50 nM, Tocris Bioscience) and DAPT (10 nM, Sigma–Aldrich), BMP7 (25 ng/ml Peprotech) and human Fgf19 (100 ng/ml, R&D). Cells were maintained in DM for 10 days. During the last 3 days DM was also supplemented with dexamethasone (3 μM). Medium was changed every 2 days.

**Immunohistochemical analysis of human organoids.** Human organoids were fixed in 10% neutral buffered formalin, washed with PBS, dehydrated, and embedded in paraffin. Five-micrometre thick sections were made from paraffin-embedded samples and sections were stained with H&E and PAS.

**Immunofluorescence analysis.** Liver organoids were extracted from Matrigel (with Cell Recovery Solution, Corning) or PEG gels (with 1 mg ml− 1 Dispase (Gibco) for 10 min at 37 °C) and fixed with 4% paraformaldehyde (PFA) in PBS (20 min, room temperature). Organoids in suspension were centrifuged (1000 r.p.m., 5 min) to remove the PFA, washed with ultrapure water and pelleted. The organoids were then spread on glass slides and allowed to attach by drying. Attached organoids were rehydrated with PBS and permeabilized with 0.2% Triton X-100 in PBS (1 h, room temperature) and blocked (1% BSA in PBS) for 1 h. Samples were then incubated overnight with phalloidin-Alexa 488 (Invitrogen) and primary antibodies against Epcam (1:50, eBioscience, G8.8), Krt19 (1:100, Abcam, ab15463), E-cadherin (1:100, Cell Signaling, 24E10), Sox9 (1:50, Millipore, AB5535), Albumin (1:50, R&D systems, MAB1455), Hnf4α (1:50, Santa Cruz, C19), ZO-1 (1:50, Invitrogen, 61-7300); YAP (1:50, Cell Signaling, 4912 S). Samples were washed with PBS and incubated for 3 h with secondary antibodies Alexa 488 donkey-α-rabbit, Alexa 568 donkey-α-mouse, Alexa 647 donkey-α-goat (1:1000 in blocking solution; Invitrogen). Following extensive washing, stained organoids were imaged by confocal (Zeiss LSM 710) mode. Dapi was used to stain nuclei.

**Western blotting.** Samples were lysed in lysis buffer (50 mM Tris (pH 7.4), 150 mM KCl, 1 mM EDTA, 1% NP-40, 5 mM NAM, 1 mM sodium butyrate, protease and phosphatase inhibitors). Proteins were separated by SDS–PAGE and transferred onto nitrocellulose or polyvinylidene difluoride membranes. Blocking (30 min) and antibody incubations (overnight) were performed in 5% BSA in TBST. YAP1 (Santa Cruz sc101199, 1:1000), phospho-YAP1[Y357] (Abcam ab62751, 1:1000), ACTIN (Santa Cruz sc47778, 1:1000), CXCL10 (RD system AF-466-NA, 1:1000), CCL2 (Novus Biologicals NBP2-22115, 1:1000), COL1A2 (Santa Cruz, 1:1000).

**Quantitative real-time qRT-PCR for mRNA quantification.** Liver organoids were extracted from Matrigel or PEG gels as previously described[12]. RNA was extracted from organoids using the RNAqueous total RNA isolation kit (Thermo Fisher) following manufacturer's instructions. RNA was transcribed to complementary DNA using QuantiTect Reverse Transcription Kit (Qiagen) following manufacturer's instructions. Expression of selected genes was analysed using the LightCycler 480 System (Roche) and SYBR Green chemistry. Quantitative reverse transcription polymerase chain reaction (PCR) results were presented relative to the mean of *Gapdh* (ΔΔCt method). Expression values shown as heatmap are reported as delta cycle threshold (ΔCt) values normalized using *Gapdh* (ΔCt values were calculated as ΔCt = Ct[target gene] − Ct[*Gapdh*] and represented by scale colour of ΔCt values [Green-low expression; Red-high expression])[51]. Primers for qRT-PCR are listed in Supplementary Table 1.

**Proliferation assay**. Cell proliferation was assessed by EdU assay (Click-iT EdU Alexa Fluor 647) following manufacturer's instructions. Liver organoids were incubated with EdU for 2 h.

For quantification of EdU+ cells, one section per organoid showing the largest dimension of the organoid was analyzed (15 organoid per condition) and from each section EdU+ cells were manually counted and expressed as percentage of cells calculated from nuclear labeling with DAPI.

**Functional analysis**. LDL uptake was detected with DiI-Ac-LDL (Biomedical Technologies). Mouse albumin secretion was detected with ELISA kit (Abcam, ab108792). Urea secretion was assessed with QuantiChrom™ Urea Assay Kit (BioAssay Systems). All experiments were performed according to the manufacturers' instructions.

**Statistical analysis and sample information**. Statistically significant differences between the means of two groups were assessed as specified in the legends. All statistical analyses were performed in the GraphPad Prism 7.0 software. A $P$-value <0.05 was considered statistically significant.

**Reporting summary**. Further information on research design is available in the Nature Research Reporting Summary linked to this article.

## Data availability

All summary or representative data generated and supporting the findings of this study are available within the paper. The data underlying Figs. 1b, d, 2a, e, f, 3a, c, e, f, 4a, c–f, 5a and Supplementary Figs. 1b, 2a–c, 3a–c, g, 4c and uncropped blots are provided as a Source Data file. Source data are provided with this manuscript.

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

## Acknowledgements

We thank Andréane Fouassier, Sabrina Bichet, Thibaud Clerc, Laure Vogeleisen-Delpech, Fabiana Fraga, the Phenotyping Unit (UDP) and the Histology core facility (HCF) of EPFL for technical assistance. The work of K.S. was funded by the Swiss National Science Foundation (SNSF 31003A_166695), Sinergia CRSII3_160798/1, the Kristian Gerhard Jebsen Foundation, and the Ecole Polytechnique Fédérale de Lausanne (EPFL). The work of M.P.L. in the area of organoid biology and technology was supported by the Swiss National Science Foundation (grant #310030_179447), the European Union's Horizon 2020 research and innovation programme (INTENS 668294), the Personalized Health and Related Technologies Initiative from the ETH Board, the Vienna Science and Technology Fund and École Polytechnique Fédérale de Lausanne (EPFL). G.S. was funded by a postdoctoral FEBS long-term fellowship.

## Author contributions

G.S., S.R., M.L. and K.S. conceived the project and wrote the manuscript. G.S., S.R. and E.Y. planned and performed experiments and analysed data. S.N. and M.H. provided human biopsy samples and critically revised the manuscript.

## Competing interests

Ecole Polytechnique Fédérale de Lausanne has filed patent applications pertaining to synthetic gels for epithelial stem cell and organoid cultures (with M.P.L.), as well as liver disease modelling (with K.S., M.P.L., G.S., E.Y. and S.R.). The remaining authors declare no competing interests.
