## [Peer Review File · Nature Communications]

Reviewers' comments:

Reviewer #1 (Remarks to the Author):

One of corresponding authors, Dr. Lutolf, previously reported chemically defined matrices for intestinal stem cell culture and organoid derivation (ref 7, Nature 2016). In this study (NCOMMS-19-29398), the authors did the similar experiments using chemically defined hydrogels (inert poly(ethylene glycol), PEG) and hepatic cells. The authors insist that they succeeded to prepare mouse and human liver organoids through actomyosin-independent YAP activation. The aim of this study is clear, however, their results are not enough to conclude liver organoid derivation and its molecular mechanism. Therefore, I can not recommend the present manuscript for publication.

Major comments

1. The liver is composed of several types of cells including hepatocyte, stellate, biliary, and Kupffer cells etc. So, the authors' definition of liver organoid is not enough. The authors should show cell-types of the organoid by FACS analysis (Ouchi et al. Cell Metabolism 2019).
2. In Fig. 3, the authors used only one inhibitor, blebbistain, as acto-myosin contractility inhibition. The authors should use another inhibitor, ROCK inhibitor Y-27632.
3. Dasatinib inhibits Src family member kinases, which affect actin cytoskeleton. The authors should explain the pathway(s) of the actomyosin-independent YAP activation in detail.

Reviewer #2 (Remarks to the Author):

Sorrentino et al. present results on a synthetic poly(ethylene glycol) (PEG) hydrogel for the derivation of both mouse and human hepatic organoids. This elegant study demonstrates equivalent hepatic organoid generation between the synthetic hydrogel and Matrigel. Furthermore, this synthetic system overcomes major limitations of tumor-derived Matrigel and represents a significant technical advance. The authors also show that the stiffness of PEG hydrogel modulates hepatic organoid development and provide results implicating Src-family kinases and YAP in this response. Results for fibrosis-associated genes in stiffer hydrogels are presented, but this data is rather preliminary and not particularly compelling.

1. Are the results unique for PEG hydrogels? Do hepatic organoids form in alginate (with/without RGD peptides) or collagen gels?
2. Similar levels of organoid formation and differentiation are observed among hydrogels presenting laminin-1, fibronectin, and RGD peptide (Fig. 1b, d). Are the densities of these adhesive motifs equivalent across the groups? Do the organoids deposit an extracellular matrix at the hydrogel interface?
3. Immunostaining for hepatic markers and differentiation should be assessed for the murine organoids established without Matrigel culture (Fig. 1f, S1).
4. Fig. 3a shows differences in organoid formation efficiency as a function of hydrogel stiffness. The authors should include results for proliferation and immunostaining analyses for hepatic differentiation markers to more fully characterize these observations. Furthermore, the stiffness of Matrigel should be reported and the results discussed in this context.
5. Results for fibrosis-associated genes in stiffer hydrogels are interesting (Fig. 4), but this data is rather preliminary and not particularly compelling. Results at protein level and matrix deposition should be included to support the gene expression data.

6. The statement "abnormal ECM stiffness is sufficient to decrease the liver progenitor proliferative capacity" (line 161) should be supported with data.

7. For the results with freshly isolated liver biopsies from patients (without Matrigel, Fig. 5f), immunostained images for hepatic markers should be presented and compared to those grown in Matrigel.

Reviewer #3 (Remarks to the Author):

In this manuscript the authors report chemically defined hydrogels (PEG-RGD) for the efficient derivation of both mouse and human hepatic organoids. They show these organoids can be readily differentiated into hepatocyte-like cells. Moreover, they identify Src family of kinases (SFKs) rather than actomyosin contractility, mediated yes-associated protein 1 (YAP) activity that is required for hepatic organoid growth. Finally, they generate clinically relevant human hepatic organoids from human patients with PEG-RGD. The manuscript is clear and well-written. However, there are important issues that should be addressed to improve the manuscript:

1. Although PEG-RGD enable to support the growth of hepatic organoids, the long-term expansion potential of these organoids is not clear. The authors only show the relevant data of passage 3 (figure 1f). There is no additional experiments, which convincingly shows that the PEG-RGD-derived organoids can be expanded for an extended period of time. After all, hepatic organoids in MATRIGEL could expand for more than 6 months with a consistent doubling time in a previous study (PMID: 25533785).

2. The authors should deeper analyze the identity of organoids. Comparing the whole genome transcriptome of PEG-RGD-derived organoids with the MATRIGEL-derived organoids would be needed.

3. Regarding the analysis of hepatocyte differentiation capacity of PEG-RGD-derived organoids, the authors should include normal human/mouse hepatocytes as positive controls (adult and fetal if possible) in their gene expression assays and in functional assays (figure 2a, 2e, 2f, figure s2b and s4b). Without these controls, the data presented has limited significance.

4. After differentiation, the PEG-RGD-derived organoids seemingly show less ALB expression than MATRIGEL-derived organoids, evidenced by immunostaining (figure 2b). Does this mean that the PEG-RGD-derived organoids have compromised hepatocyte differentiation potential? Moreover, the organoids in expansion medium already express Ecad and HNF4A before differentiation. Therefore, these markers are not appropriate for assessing the extent of hepatocyte differentiation. To clarify this situation, quantitative data along with the immunostainings should be provided in figure 2b-2d. Additionally, the analysis of hepatocyte differentiation of MATRIGEL-derived organoids should be as a control in figure 2 and figure s2.

5. The authors demonstrate that organoid formation and growth in PEG-RGD depend on YAP activation. Whether the YAP activators such as lysophosphatidic acid (LPA) and sphingosine-1-phosphate (S1P) could improve the organoid formation efficiency in PEG-RGD of various stiffness?

6. Increased stiffness (4kb) result in decreased organoid proliferation without affecting YAP activation, suggesting the existence of other pathways controlling liver organoid growth in response to increased stiffness. Recently, Aloia L et al demonstrated that both of YAP-Hippo and ErbB-MAPK signaling pathways play crucial role in adult cholangiocyte-derived organoid formation (PMID: 31685987). Are ErbB-MAPK signaling involved in regulation of organoid formation in aberrant stiffness (4kb)? This needs to be further addressed.

7. Finally, the authors show that clinically relevant human stem/progenitor cells can be grown in vitro without any requirement of animal-derived matrices and state that this may open exciting perspectives for liver organoid-based clinical applications. Because large number of cells are need for clinical applications, it would be good to show the expansion potential of these biopsy-derived human liver organoids. Besides, whether these organoids could be frozen and thawed is also critical. Karyotype analysis of biopsy-derived human organoids in PEG-RGD of early and later passages should be provided.

8. In figure 5c and 5d, please exchange the label PEG-RGD (HDM) with PEG-RGD (HEM).

Point by point response to the referees' comments

We would like to thank the editor for giving us the opportunity to submit a revised version of our manuscript as well as the three reviewers for their constructive suggestions. In this new version of our manuscript we have addressed all the major points raised by the reviewers. All the relevant new findings have now been integrated in the manuscript and corroborate and significantly strengthen our initial data.

Please find below the point-by-point rebuttal to all reviewers' comments:

Reviewer 1 comments:

One of corresponding authors, Dr. Lutolf, previously reported chemically defined matrices for intestinal stem cell culture and organoid derivation (ref 7, Nature 2016). In this study (NCOMMS-19-29398), the authors did the similar experiments using chemically defined hydrogels (inert poly(ethylene glycol), PEG) and hepatic cells. The authors insist that they succeeded to prepare mouse and human liver organoids through actomyosin-independent YAP activation. The aim of this study is clear, however, their results are not enough to conclude liver organoid derivation and its molecular mechanism. Therefore, I can not recommend the present manuscript for publication.

Although we previously reported a method to derive and culture intestinal organoids in defined matrices (*Gjorevski et al. Nature 2016*), we respectfully disagree with the statement that we conducted "similar experiments" using hepatic cells. On the contrary, we would like to emphasize that there are major differences in scope and potential clinical impact between both studies:

1. Clinical relevance:

In *Gjorevski et al.* we used laminin-containing PEG hydrogels for culturing human intestinal organoids. Importantly, the laminin used in the study was animal-derived, and as such the materials were clinically incompatible. In our current study, instead, we have overcome this limitation by using materials that are entirely free of any animal-derived product at any step of the procedure, and thus clinically applicable. To the best of our knowledge, the methodology that we show in this manuscript is the first one to yield liver organoids in a manner that is truly compatible with clinical applications.

2. Tissue mechanics modeling and insight into stiffness-related mechanisms of diseases:

In *Gjorevski et al.* we did not test whether PEG hydrogels could be used to model key aspects of disease. In this manuscript, instead, we were able to model in 3D cultures some features (i.e. stiffness) of the aberrant tissue mechanics that characterize the fibrotic liver. By using this approach, we demonstrated for the first time that abnormal tissue stiffness has, *per se*, a negative impact on liver progenitor cell fitness.

3. Mechanistic insights on stiffness-related phenotypes

In contrast to the data published in *Gjorevski et al.*, our current study provides unequivocal evidence that in case of liver progenitor cells, the role of ECM stiffness on proliferation is largely actomyosin-independent and reveals the existence of a non-canonical pathway in which YAP transcriptional activation requires phosphorylation of YAP by the Src tyrosine kinase.

Based on these novel findings, we are convinced that our study represents an important conceptual advance to the field of liver organoids.

Major comments

1. The liver is composed of several types of cells including hepatocyte, stellate, biliary, and Kupffer cells etc. So, the authors' definition of liver organoid is not enough. The authors should show cell-types of the organoid by FACS analysis (Ouchi et al. Cell Metabolism 2019).

We agree that the nomenclature in the field of organoids can be confusing and we thank this reviewer for giving us the opportunity to clarify this point.

Liver organoids can be derived from both iPSC (*Takebe et al. Cell Reports 2017; Takebe et al. Nature 2013; Ouchi et al. Cell Stem Cell 2019*) and adult bi-potent stem cells (also known as progenitors) (*Huch et al. Nature 2012; Huch et al. Cell 2015; Planas-Paz et al. Cell Stem Cell 2019; Pepe-Mooney et al. Cell Stem Cell 2019; Aloia et al. Nature Cell Biology 2019*). In the first case, organoids are comprised of various liver cells (e.g. hepatocytes, cholangiocytes, stellate cells, and Kupffer cells), however, iPSC-derived organoids could be potentially tumorigenic and are thus suboptimal for clinical applications (*Gutierrez-Aranda et al., Stem Cells 2010*). In the second case, the organoids generated from adult stem cells are homogeneously composed of hepatic progenitor cells that can be eventually differentiated into cholangiocytes or hepatocyte-like cells after modification of the culture medium. These organoids are genetically stable and not tumorigenic (*Huch et al. Cell 2015*), therefore potentially relevant for regenerative therapy purposes. We have used the latter system and are hence in line with the definition and nomenclature of liver organoids as it is currently used in the field.

In this context it is important to underscore that the choice of using adult stem cell-derived organoids here in this study was not to generate a culture system that recapitulates the liver in its entity, but rather to generate cultures of hepatic progenitor cells suitable for stem-cell-based regenerative medicine. Indeed, in the case of patients with liver failure (which is a direct consequence of the collapse of hepatocyte function), the ultimate goal of liver regenerative medicine is to repopulate the liver parenchyma with exogenously-derived functional hepatocytes to rescue liver function.

Based on these premises, we do not expect any other cell type appearing in our organoid cultures. To confirm this, we performed qRT-PCR analysis for markers of different liver cell types. As shown in the new **Supplementary Fig. 1b**, the results indicate that only progenitor/stem cell markers are expressed in the organoid culture.

2. In Fig. 3, the authors used only one inhibitor, blebbistatin, as acto-myosin contractility inhibition. The authors should use another inhibitor, ROCK inhibitor Y-27632.

We thank the reviewer for suggesting this experiment. As shown in the new **Supplementary Figure 3f** we performed a new experiment in PEG hydrogels with physiological stiffness in which we tested the impact on organoid growth of two different ROCK inhibitors (Y27632 and Thiazovivin) and compared those with Blebbistatin. Quantification of organoid formation efficiency 3 days after cell seeding demonstrated that similar to Blebbistatin, the two ROCK inhibitors have an obvious stimulatory effect on organoid growth.

3. Dasatinib inhibits Src family member kinases, which affect actin cytoskeleton. The authors should explain the pathway(s) of the actomyosin-independent YAP activation in detail.

Although Src has an important role as an upstream regulator of actomyosin contraction and actomyosin-dependent YAP activation, our data undoubtedly demonstrate that the effects of Src inhibition (by Dasatinib) on organoid growth are incompatible with a mechanism in which Src controls organoid growth by promoting actomyosin contraction. Indeed, inhibition of actomyosin contraction obtained by treating organoids with the most established actomyosin inhibitor (i.e. Blebbistatin) resulted in a clear improvement of organoid growth. Our results are instead in line with a mechanism in which Src directly phosphorylates and activates YAP in response to increased ECM stiffness (Supplementary Figure 3e). Indeed, Src can directly phosphorylate YAP on a stimulatory tyrosine residue upon activation by mechanotransducers such as Integrin $\beta 4$ or focal adhesion kinase (*Kim, N. et al. J. Cell Biol. 2015; Li, P. et al. Genes Dev. 2016*), or indirectly through the activation of the Hippo pathway upstream

regulators MST1/2 (Lamar et al. JBC 2019). This event occurs in the plasma membrane and is independent of actomyosin contraction. The direct phosphorylation of YAP by Src is well established in the field and it is supported by extensive literature [Taniguchi, K. et al. *Nature* (2015); Calvo, F. et al. *Nat Cell Biol* (2013); Smoot, R. et al. *J. Cell.Biochem.* (2018); Kim, N. et al. *J. Cell Biol.* (2015); Sugihara, T. et al. *Mol. Cancer Res.* (2018); Tamm, C.J. *Cell Sci.* (2011); Li, P. et al. *Genes Dev.* (2016)]. The studies relevant to Src-dependent YAP activation are now integrated in the new version of our manuscript.

Reviewer 2 comments:

Sorrentino et al. present results on a synthetic poly(ethylene glycol) (PEG) hydrogel for the derivation of both mouse and human hepatic organoids. This elegant study demonstrates equivalent hepatic organoid generation between the synthetic hydrogel and Matrigel. Furthermore, this synthetic system overcomes major limitations of tumor-derived Matrigel and represents a significant technical advance. The authors also show that the stiffness of PEG hydrogel modulates hepatic organoid development and provide results implicating Src-family kinases and YAP in this response. Results for fibrosis-associated genes in stiffer hydrogels are presented, but this data is rather preliminary and not particularly compelling.

We thank the reviewer to find our study elegant and to state that it represents a significant technical advance. About fibrosis-associated genes please see reply to point 5.

1. Are the results unique for PEG hydrogels? Do hepatic organoids form in alginate (with/without RGD peptides) or collagen gels?

To address whether liver organoids can be derived within natural-derived matrices such as collagen and alginate, we selected collagen as proof-of-principle and performed a new experiment in which we encapsulated liver progenitor cells into both Matrigel and Type I collagen gel. As depicted in **Figure A**, liver organoids could be derived in collagen gels, although with less efficiency compared to Matrigel. While this is interesting, we think that these results are beyond the scope of our work. Natural matrices suffer from some

Figure A: Representative picture of mouse liver progenitor cells 6 days after embedding in 10µL droplet of Matrigel or Type I Collagen (8-11 mg/ml prepared following manufacturer's instruction – from Corning).

important drawbacks, such as low stiffness, limited long-term stability, batch-to-batch variability and are likely not conducive to controlled modifications. Therefore, both collagen- and alginate-based gels are not ideal for clinical application when compared to synthetic materials such as PEG. Based on these facts, we decided not to include these results in the new version of the manuscript.

2. Similar levels of organoid formation and differentiation are observed among hydrogels presenting laminin-1, fibronectin, and RGD peptide (Fig. 1b, d). Are the densities of these adhesive motifs equivalent across the groups?

The main goal of this experiment was to demonstrate that inclusion of a minimal RGD motif is sufficient to render the PEG bioactive in such a way that it allows organoid derivation and expansion with an efficiency comparable to Matrigel. Therefore, the molar concentration of the ECM proteins was not chosen to be similar between conditions. In particular, we used fibronectin, laminin-1 and collagen-IV at 0.5, 0.2 and 0.5 mg/ml respectively. We have now highlighted this information in the M&M section.

Do the organoids deposit an extracellular matrix at the hydrogel interface?

This is an intriguing question. To address this point, we assessed ECM protein deposition in the hydrogel/organoid interface in conditions that maintain the native structure of the hydrogel system. We performed immunofluorescence staining without extracting the organoid from the PEG hydrogel following a protocol that we have recently described (*Gjorevski et al. Nature Protoc. 2017*). In line with published data (*Vyas et al. Hepatology 2018*), the experiment confirmed that liver progenitor cells were not able to deposit ECM proteins at least not to an amount that is detectable by immunofluorescence (data not shown).

3. Immunostaining for hepatic markers and differentiation should be assessed for the murine organoids established without Matrigel culture (Fig. 1f, S1).

The reviewer is correct. To address this point, we directly embedded freshly isolated mouse bile ducts in PEG-RGD hydrogels and after the first passage, we started the differentiation. As shown in the new **Supplementary Fig. 2b**, organoids directly derived in PEG-RGD could be efficiently differentiated into hepatocyte-like cells with no significant differences compared to organoids generated and cultured in Matrigel. Moreover, markers of differentiated hepatocytes (i.e. Albumin (ALB) and Glutamine synthase (GLUL)) showed a similar protein expression pattern in organoids differentiated in PEG and Matrigel (new **Supplementary Fig. 2c**).

4. Fig. 3a shows differences in organoid formation efficiency as a function of hydrogel stiffness. The authors should include results for proliferation and immunostaining analyses for hepatic differentiation markers to more fully characterize these observations.

We thank the reviewer for this suggestion. To address this point, we performed EdU assay and quantified the number of proliferative cells in organoids cultured in expansion medium for 6 days in PEG-RGD hydrogels with different stiffness. As shown in the new **Supplementary Fig. 3a**, we found that physiological stiffness was associated with increased proliferation compared to soft and fibrosis-like stiffness.

Regarding the question on differentiation markers, we already showed in the original version of the manuscript (**Supplementary Figure 3b**) that the stiffness of ECM did not impact on hepatocyte differentiation.

Furthermore, the stiffness of Matrigel should be reported and the results discussed in this context.

Discrepancies in the mechanical properties of Matrigel have been reported in the literature. These are due to inherent variability between batches and within a single batch (*Reed J et al. 2009; Soofi SS et al 2009*). Variabilities are also attributed to different testing methods (bulk vs AFM) and temperature (*Soofi SS et al 2009; Alcaraz J et al. 2008*). Therefore, the stiffness of the Matrigel is not an absolute property, rather it depends on many additional variables.

We think that a comparison between Matrigel and PEG stiffness would not be relevant as Matrigel has beyond stiffness, several uncontrolled biochemical features (unknown composition, unknown concentration of different ECM proteins, presence of growth factors...) that make it impossible to have a meaningful comparison with a fully defined system such as PEG. Therefore, we respectfully believe that any discussion about Matrigel and PEG stiffness comparison in our manuscript might be confounding and would not add any relevant information to our study.

5. Results for fibrosis-associated genes in stiffer hydrogels are interesting (Fig. 4), but this data is rather preliminary and not particularly compelling. Results at protein level and matrix deposition should be included to support the gene expression data.

Concerning this question, we would like to clarify that in the current study we do not want to claim that fibrosis-mimicking hydrogels induce the expression of “fibrosis-associated genes” and that organoids contribute to matrix deposition; instead we stated on page 6 “*In this condition liver organoids showed a reduction in the expression of hepatic progenitor markers (Fig. 4c) and upregulation of genes involved in cellular response to hepatic injury*

(Fig. 4d), indicating impaired stemness potential and concomitant induction of a stress response". To support this, we showed upregulation of stress-associated genes, including *Crp*, and chemokines/cytokines, such as *Tnf α* , *Tgf β* and *Ccl2*, which are classically induced in hepatocytes to recruit immune cells to the site of damage. Even though we observed an increase in *Colla2*, we did not see any substantial matrix deposition on the interface between organoids and hydrogel (please refer to point 2).

Induction of genes involved in acute stress response, at the mRNA level, is the most used readout to monitor hepatic stress, therefore we think we are consistent with the experimental approach broadly used in the field. However, as suggested by the reviewer and to corroborate our observation, we now performed a new Western blot experiment in the same condition used for **Figure 4d** and confirmed that increased ECM stiffness induced the expression of CCL2 at the protein level (new **Supplementary Fig. 4a**). Moreover, we found that fibrosis-like ECM increased the mRNA and protein levels of *Cxcl10*, another key chemokine known to be secreted by hepatic epithelial cells, and we included these results in the **Figure 4d** and in the new **Supplementary Figure 4a**. Along with these results, we confirmed the upregulation of COL1A2 at the protein level (new **Supplementary Figure 4a**). These results are in line with published results showing that liver epithelial cells in normal and fibrotic rat and human liver tissue (from patients with a variety of liver diseases including viral, alcoholic, and biliary) were found to express mRNA for different ECM proteins including collagens (*Rescan PY et al. Am J Pathol. 1993*; *Xia JL et al. Am J Pathol. 2006*). Although our new experiment showing no ECM deposition by organoids (see point 2) suggests that the increase in COL1A2 expression is most likely not sufficient to contribute to a fibrotic phenotype, the increased expression of collagen mRNA further corroborates the stiffness-mediated induction of a stress response in liver progenitor cells.

Although we do not consider these data as the primary discovery in our manuscript (i.e. the identification of a clinical-grade matrix system for liver organoids), we believe that the results in Figure 4 are highly interesting as they reveal for the first time that liver progenitor cells can sense the surrounding physical microenvironment and respond to aberrant physical cues by activating an adaptive cellular response. In particular, the demonstration that ECM stiffness increases chemokine expression at the mRNA and protein levels represents an important discovery on how tissue mechanics may contribute to orchestrating the liver immune response during liver fibrosis.

6. The statement "abnormal ECM stiffness is sufficient to decrease the liver progenitor proliferative capacity" (line 161) should be supported with data.

Please see reply to point 4.

7. For the results with freshly isolated liver biopsies from patients (without Matrigel, Fig. 5f), immunostained images for hepatic markers should be presented and compared to those grown in Matrigel.

We agree with the reviewer. To address this point, we directly embedded a freshly isolated human biopsy in Matrigel or in PEG hydrogel and immediately after organoid formation we induced their differentiation. As shown in the new **Supplementary Fig. 4d**, we found cells positive for hepatocyte markers (i.e. Albumin (ALB) and Glutamine synthetase (GLUL)) in organoids generated in Matrigel and PEG.

Reviewer 3 comments:

In this manuscript the authors report chemically defined hydrogels (PEG-RGD) for the efficient derivation of both mouse and human hepatic organoids. They show these organoids can be readily differentiated into hepatocyte-like cells. Moreover, they identify Src family of kinases (SFKs) rather than actomyosin contractility, mediated yes-associated protein 1 (YAP) activity that is required for hepatic organoid growth. Finally, they generate clinically relevant human hepatic organoids from human patients with PEG-RGD. The manuscript is clear and well-written. However, there are important issues that should be addressed to improve the manuscript:

We thank the reviewer for considering our manuscript clear and well written.

1. Although PEG-RGD enable to support the growth of hepatic organoids, the long-term expansion potential of these organoids is not clear. The authors only show the relevant data of passage 3 (figure 1f). There is no additional experiments, which convincingly shows that the PEG-RGD-derived organoids can be expanded for an extended period of time. After all, hepatic organoids in MATRIGEL could expand for more than 6 months with a consistent doubling time in a previous study (PMID: 25533785).

We received comments from Reviewer 3 on 21st January 2020 and immediately set up an experiment to show that organoids in PEG hydrogel can be expanded for an extended period of time (more than 3 passages). To this aim, mouse liver organoids were embedded in PEG hydrogel and serially passaged by dissociation to single cells (at 1 to 3 dilution ratio) for about 2 months (which corresponds to 6 passages). As shown in **new Supplementary Figure 1c**, after 2 months in culture, organoids were still fully capable to grow and reach the confluence in 7 days after single cell seeding. At this point, unfortunately we had to stop this long-term experiment due to the lab shutdown caused by COVID19 pandemic, but we hope that the reviewer will be reassured by the proof-of-principle that after 2 months in culture in PEG hydrogels, liver organoids are still able to grow with the same efficiency.

2. The authors should deeper analyze the identity of organoids. Comparing the whole genome transcriptome of PEG-RGD-derived organoids with the MATRIGEL-derived organoids would be needed.

Matrigel is not a clinical grade material, and as such, should not be considered as a benchmark for the culture system that we show in our manuscript. Therefore, we strongly believe that any comprehensive comparison between Matrigel and PEG would not bring any additional information to our work. However, we do agree that the analysis of cell identity in organoids grown in PEG hydrogels is important to evaluate the capacity of PEG hydrogels to maintain a homogeneous culture of progenitor cells and to prevent spontaneous trans-differentiation into other cell types (e.g. stromal cells) that could have a detrimental impact on both safety and reproducibility. To confirm that PEG matrices maintain a homogeneous culture of liver progenitor cells, we performed qPCR analysis for markers of different liver cell types. As shown in the new **Supplementary Fig. 1b**, the results confirm that both in Matrigel and PEG, the organoid cultures are composed of only liver progenitor cells.

3. Regarding the analysis of hepatocyte differentiation capacity of PEG-RGD-derived organoids, the authors should include normal human/mouse hepatocytes as positive controls (adult and fetal if possible) in their gene expression assays and in functional assays (figure 2a, 2e, 2f, figure s2b and s4b). Without these controls, the data presented has limited significance.

Liver organoids have been extensively compared with isolated hepatocytes or liver tissue in many previous studies (for example: Hu et al. Cell 2018; Huch et al. Cell 2015; Huch et al. Nature 2013). These papers showed that liver organoids grown in Matrigel are less functional than freshly isolated hepatocytes, and there is an overall agreement in the field that the currently available protocol would benefit from adaptation and improvement.

In the current study, we do not claim that our system improves the hepatic identity of liver organoids as compared to Matrigel. Instead, following the gold standard protocol used in the field, we provide evidence that organoids cultured in PEG hydrogels maintain their potential to differentiate into hepatocyte-like cells.

To address the reviewers' point, we compared the expression of hepatocyte markers in organoids differentiated in PEG hydrogels or Matrigel with those of freshly isolated, and immediately harvested hepatocytes. As shown in **Figure B** below, although the expression of several markers was comparable between fresh hepatocytes and organoids in both PEG and Matrigel, some key hepatocyte-specific genes such as Albumin and Cyp3a11 were significantly less expressed in organoids, as was shown already in the literature. Further, we compared the differentiation capacity of organoids cultured in PEG with those cultured in Matrigel. Organoids grown in PEG-

RGD could be efficiently differentiated into hepatocyte-like cells with no significant differences compared to organoids generated and cultured in Matrigel (new **Supplementary Fig. 2b**).

In sum, although we are strongly convinced that the differentiation process needs to be improved in both Matrigel and PEG, our results demonstrate that liver organoids grown in PEG preserve their potential to be differentiated into hepatocyte-like cells.

Figure B: mRNA levels of indicated genes were measured by qRT-PCR in freshly isolated primary hepatocytes or in liver organoids cultured in differentiation medium in PEG-RGD hydrogels or Matrigel (MG). Expression values shown as heatmap are reported as delta cycle threshold (ΔCt) values normalized using *Gapdh* (ΔCt values were calculated as $\Delta Ct = Ct[\text{target gene}] - Ct[\text{Gapdh}]$ and represented by scale colour of ΔCt values [Green-low expression; Red-high expression]).

4. After differentiation, the PEG-RGD-derived organoids seemingly show less ALB expression than MATRIGEL-derived organoids, evidenced by immunostaining (figure 2b). Does this mean that the PEG-RGD-derived organoids have compromised hepatocyte differentiation potential?

Moreover, the organoids in expansion medium already express *Ecad* and *HNF4A* before differentiation. Therefore, these markers are not appropriate for assessing the extent of hepatocyte differentiation. To clarify this situation, quantitative data along with the immunostainings should be provided in figure 2b-2d. Additionally, the analysis of hepatocyte differentiation of MATRIGEL-derived organoids should be as a control in figure 2 and figure s2.

To address these points and to quantitatively compare differentiation in Matrigel and PEG gels we performed a new experiment in which we observed no significant differences in the induction of hepatocyte marker expression (**Supplementary Fig. 2b**). Moreover, in addition to Albumin (ALB), we included a new staining of Glutamine synthetase (GLUL), a hepatocyte marker that is significantly induced after differentiation (**Supplementary Figures 2e and 4d**).

5. The authors demonstrate that organoid formation and growth in PEG-RGD depend on YAP activation. Whether the YAP activators such as lysophosphatidic acid (LPA) and sphingosine-1-phosphate (S1P) could improve the organoid formation efficiency in PEG-RGD of various stiffness?

LPA and S1P control YAP activation via RhoA-actin (Plouffe et al Mol Cell 2016; Yu et al, Cell 2012). In the original version of our manuscript, we indicated that the RhoA-actin axis is not only dispensable, but even detrimental for liver organoid growth in PEG hydrogel with physiological stiffness. In our revised manuscript, we have significantly strengthened this claim by showing that RhoA inhibition (using two different compounds), as well as actomyosin contraction inhibition, sustain liver organoid growth (**new Supplementary Figure 3f**).

The reviewer is asking whether known YAP activators such as LPA and S1P, could rescue organoid growth in soft matrices. To address this question, progenitor cells encapsulated in soft PEG-RGD hydrogels were treated

with LPA or S1P for 6 days (as in Yu et al, Cell 2012). As shown in **Figure C** below, treatment with LPA or S1P could not rescue organoid growth in soft hydrogels and did not impact on YAP phosphorylation. The lack of effect on YAP phosphorylation is intriguing and could be related to the use of our 3D culture system. LPA and S1P are known YAP upstream activators that promote YAP-mediated proliferation and migration in cells grown in 2D cultures (Yu et al, Cell 2012). Different studies have shown that LPA and S1P control YAP by promoting RhoA activation and consequent acto-myosin contraction (Plouffe et al Mol Cell 2016; Yu et al, Cell 2012). These published data, together with our evidence showing that organoid growth in 3D PEG hydrogels is actomyosin-independent (Figure 3), strongly indicate that the RhoA-actin axis is not involved in promoting liver organoid growth in PEG hydrogel.

Figure C: Upper panel: Representative picture of mouse liver progenitor cells 6 days after embedding in PEG-RGD hydrogels with normal (1.3 kPa) or soft (0.3 kPa) stiffness and treated with LPA (10 μM) and S1P (10 μM) for 6 days. Treatments were replenished every two days. Vehicle is water (for S1P) or water/ethanol (for LPA). Lower panel: liver organoids cultured in PEG-RGD with soft (0.3 kPa) stiffness were treated with LPA (10 μM) and S1P (10 μM) for 1 hour. Cell lysates were subjected to immunoblotting with the indicated antibodies: YAP1 (Santa Cruz sc101199, 1:1000), phospho-YAP1 (Cell Signaling 4911, 1:1000), ACTIN (Santa Cruz sc47778, 1:1000).

Increased stiffness (4kb) result in decreased organoid proliferation without affecting YAP activation, suggesting the existence of other pathways controlling liver organoid growth in response to increased

stiffness. Recently, Aloia L et al demonstrated that both of YAP–Hippo and ErbB–MAPK signaling pathways play crucial role in adult cholangiocyte-derived organoid formation (PMID: 31685987). Are ErbB–MAPK signaling involved in regulation of organoid formation in aberrant stiffness (4kb)? This needs to be further addressed.

This is an interesting suggestion. To assess whether the reduction of organoid growth in the fibrosis-like stiff hydrogel is associated with modulation of the ErbB-MAPK pathway, we decided to encapsulate progenitors in normal (1.3 kPa) or fibrosis-like (4 kPa) stiff hydrogels and monitor, in time, the expression of a panel of ErbB-MAPK downstream targets (please note that we monitored the same genes shown in Aloia et al). As shown in the **Figure D**, the expression of ErbB-MAPK targets, did not show any clear reduction in response to increased stiffness. On the contrary, at day 6, we observed a small increase in the expression of *Jun*, *Foxo3* and *Socs2*, which further confirms that fibrosis-like hydrogels do not reduce organoid growth by interfering with the ErbB-MAPK pathway.

Figure D: mRNA levels of indicated genes were measured by qRT-PCR at indicated time points in liver progenitor cells embedded in PEG-RGD hydrogels with normal (1.3 kPa) or fibrosis-like (4 kPa) stiffness. The heatmap represents fold-changes relative to day 1 of 1.3 kPa point.

7. Finally, the authors show that clinically relevant human stem/progenitor cells can be grown in vitro without any requirement of animal-derived matrices and state that this may open exciting perspectives for liver organoid-based clinical applications. Because large number of cells are need for clinical applications, it would be good to show the expansion potential of these biopsy-derived human liver organoids. Besides, whether these oragnods could be frozen and thawed is also critical. Karyotype analysis of biopsy-derived human oragnods in PEG-RGD of early and later passages should be provided.

We thank the reviewer for these suggestions and agree that the possibility of freezing/thawing organoids cultured in PEG gels is an important feature. To address this point, we have frozen human liver organoids cultured in PEG (without any Matrigel step) and, after 1 month, the cells were thawed and re-encapsulated in PEG gels. As shown in the **new Supplementary Figure 4e**, organoids can efficiently recover and grow after thawing. Unfortunately, as for the long-term culture experiment described above, the karyotype analysis has been impeded by the current COVID19 pandemic and lockdown of our campus.

We are aware that the monitoring of long-term cultures as well as the optimization of the expansion and differentiation of hepatic progenitor cells in PEG are important features that deserve comprehensive investigation. Indeed, our main goal was to provide proof-of-principle that liver organoids can be cultured in clinical-grade matrices and establish a framework to study hepatic progenitor cells in a defined mechanical environment. We agree with the reviewer that in the future, it will be important to gain further knowledge about how we can improve the conditions as to comply with therapeutic needs. Those investigations should go hand in hand with advances coming from the regenerative medicine angle. For instance, an important aspect that is still poorly defined and unfortunately remains speculative is the number of cells that is required for a regenerative therapy protocol in humans. These additional insights are essential but will require an investment that we feel is beyond the scope of the current study.

8. In figure 5c and 5d, please exchange the label PEG-RGD (HDM) with PEG-RGD (HEM).

We thank the reviewer for noticing this typo. The labels have now been corrected.

REVIEWERS' COMMENTS:

Reviewer #1 (Remarks to the Author):

The authors have responded to almost all of concerns of this reviewer by performing new experiments and incorporating more experimental data. The new information significantly strengthens a detailed study.

Reviewer #2 (Remarks to the Author):

The authors have adequately addressed several of my prior technical critiques. However, I remain concerned about other points and the response provided to the other reviewers' critiques raise significant concerns over the broad significance of the work:

1. Results and interpretation of fibrosis-associated genes in stiffer hydrogels (Fig 4) continue to be problematic. It is still not clear whether these increases in gene expression have any impact related to fibrosis. While there are no differences in matrix production by the differentiated cells, the authors should explore whether secreted factors influence the fate of other cells that play central roles in fibrosis. Furthermore, the authors' response to Reviewer 3 critique regarding the Aloia paper (comment 6) confounds interpretation of the data and weakens that authors' conclusions. It is not clear why the additional results (Fig D) are not included in the revised manuscript as these results are directly relevant to the studies performed.
2. The results with collagen gels (Fig A) should be included as supplementary materials as these findings show that the response is not unique to Matrigel or PEG gels.
3. The mechanical properties of Matrigel, measured in the same fashion as those for the PEG gels, should be reported and discussed in the manuscript. While the technical points raised by the authors are valid, their argument for not providing this information is not compelling, especially given the main point of the study regarding the influence of matrix mechanics on organoid activities.
4. Results and discussion provided in response to Reviewer 3's comment 3 (Fig B) should be included in the revised manuscript. These new results are very important and relevant in the context of the authors' conclusion regarding the differentiation of the organoids into hepatocyte differentiation. The results in Fig B reduce the significance of the study as it shows that the differentiated organoids are different from primary hepatocytes.

Reviewer #3 (Remarks to the Author):

Thanks for the author's careful response. All the questions have been well explained. I think that this manuscript could be accepted now.

REVIEWERS' COMMENTS:

Reviewer #1 (Remarks to the Author):

The authors have responded to almost all of concerns of this reviewer by performing new experiments and incorporating more experimental data. The new information significantly strengthens a detailed study.

We thank the reviewer for helping in improving the quality of our study.

Reviewer #2 (Remarks to the Author):

The authors have adequately addressed several of my prior technical critiques. However, I remain concerned about other points and the response provided to the other reviewers' critiques raise significant concerns over the broad significance of the work:

1. Results and interpretation of fibrosis-associated genes in stiffer hydrogels (Fig 4) continue to be problematic. It is still not clear whether these increases in gene expression have any impact related to fibrosis. While there are no differences in matrix production by the differentiated cells, the authors should explore whether secreted factors influence the fate of other cells that play central roles in fibrosis.

As we already emphasized in the first point-by-point rebuttal, the data shown in Figure 4 aim to demonstrate that liver progenitor cells can sense the surrounding physical microenvironment and respond to aberrant physical cues by activating an adaptive cellular response. The elucidation of the mechanisms of activation of this cellular response, as well as its functional consequences, require an effort that we believe goes beyond the scope of this study (which is focused on the identification of a clinical-grade matrix system for liver organoids) and that will be investigated in future projects in our laboratories.

Furthermore, the authors' response to Reviewer 3 critique regarding the Aloia paper (comment 6) confounds interpretation of the data and weakens that authors' conclusions. It is not clear why the additional results (Fig D) are not included in the revised manuscript as these results are directly relevant to the studies performed.

This was indeed a question raised by reviewer 3, which we did not consider as a critique, but rather as a constructive and interesting suggestion. We respectfully disagree with the interpretation of reviewer 2 that our new results in relation to the Aloia et al. paper may further confound the interpretation of our data. In particular, these data clearly show that the ErbB/Mapk pathway is not involved in the control of organoid growth by aberrant stiffness. In line with reviewer 3, who did not request the addition of these negative data in the paper, we also believe that including those data in this study would not bring essential insight or conceptual advance. Instead, we plan to include our results on ErbB/Mapk pathway in a follow-up study currently ongoing in our laboratory and specifically aimed at identifying the molecular pathways involved in the inhibition of organoid growth by aberrant stiffness.

2. The results with collagen gels (Fig A) should be included as supplementary materials as these findings show that the response is not unique to Matrigel or PEG gels.

The growth and the function of liver organoids within 3D collagen gels has been previously characterized in literature (Saheli et al. 2017, DOI: 10.1002/jcb.26622) and for this reason we did not include these data in our first revision. In order to comply with the reviewer's request, we now included the figure showing liver organoids growing in Matrigel and Collagen matrices in Supplementary Figure 1.

3. The mechanical properties of Matrigel, measured in the same fashion as those for the PEG gels, should be reported and discussed in the manuscript. While the technical points raised by the authors are valid, their argument for not providing this information is not compelling, especially given the main point of the study regarding the influence of matrix mechanics on organoid activities.

These results have been included in Supplementary Figure 3.

4. Results and discussion provided in response to Reviewer 3's comment 3 (Fig B) should be included in the revised manuscript. These new results are very important and relevant in the context of the authors' conclusion regarding the differentiation of the organoids into hepatocyte differentiation. The results in Fig B reduce the significance of the study as it shows that the differentiated organoids are different from primary hepatocytes.

As already discussed in the first point-by-point rebuttal, improving the differentiation of cholangiocyte-derived organoids is a well-known challenge in the field, and is a major focus in many labs working with liver organoids. However, we understand the reviewer's comment to highlight this current limitation in the field, which is so far unresolved. We therefore modified the figure to make it consistent and not-redundant with other panels within the figure and included those results in Supplementary Figure 2. In particular, we removed the comparison with Matrigel, which is well-described in Supplementary Figure 2c, and added comparison with undifferentiated liver organoids in order to better highlight the induction of differentiation.

Reviewer #3 (Remarks to the Author):

Thanks for the author's careful response. All the questions have been well explained. I think that this manuscript could be accepted now.

We thank the reviewer for his/her constructive comments and suggestions that significantly contributed to improve the quality of our study.